# Quantitative Analysis of Morphology and Surface Properties of Poly(lactic acid)/Poly(ε-caprolactone)/Hydrophilic Nano-Silica Blends

**DOI:** 10.3390/polym16121739

**Published:** 2024-06-19

**Authors:** Sanja Mahović Poljaček, Dino Priselac, Tamara Tomašegović, Mirela Leskovac, Aleš Šoster, Urška Stanković Elesini

**Affiliations:** 1Faculty of Graphic Arts, University of Zagreb, 10000 Zagreb, Croatia; tamara.tomasegovic@grf.unizg.hr; 2Faculty of Chemical Engineering and Technology, University of Zagreb, 10000 Zagreb, Croatia; mlesko@fkit.hr; 3Faculty of Natural Sciences and Engineering, University of Ljubljana, SI-1000 Ljubljana, Slovenia; ales.soster@ntf.uni-lj.si (A.Š.); urska.stankovic@ntf.uni-lj.si (U.S.E.)

**Keywords:** poly(lactic acid), poly(ε-caprolactone), nano-silica, morphology, adhesion parameters, EDS mapping

## Abstract

A quantitative analysis of the morphology, as well as an analysis of the distribution of components and surface/interfacial properties in poly(lactic acid)(PLA) InegoTM 3251D, poly(ε-caprolactone) (PCL) Capa 6800 and nano-silica (SiO_2_) Aerosil^®^200 blends, was conducted in this research. The study aimed to improve the understanding of how PLA, PCL, and nano-SiO_2_ interact, resulting in the specific morphology and surface properties of the blends. Samples were produced by varying the concentration of all three components. They were analyzed using SEM, EDS mapping, water contact angle measurements, surface free energy calculation, adhesion parameter measurements, and FTIR-ATR spectroscopy. The results showed that the addition of SiO_2_ nanoparticles led to an increase in the contact angle of water, making the surface more hydrophobic. SEM images of the blends showed that increasing the PCL content reduced the size of spherical PCL elements in the blends. FTIR-ATR analysis showed that SiO_2_ nanoparticles influenced the structure ordering of PLA in the blend with equal portions of PLA and PCL. In the samples with a higher PCL content, the spherical elements present in the samples with a higher PLA/PCL ratio have been reduced, indicating better interactions at the interface between PLA, PCL, and SiO_2_. SEM-EDS mapping of the PLA/PCL 100/0 blend surfaces revealed the presence of SiO_2_ clusters and the silicon (Si) concentration reaching up to ten times higher than the nominal concentration of SiO_2_. However, with the addition of 3% SiO_2_ to the blend containing PCL, the structure became more granular. Specifically, Si protrusions in the sample PLA/PCL 90/10 with 3% SiO_2_ displayed 29.25% of Si, and the sample PLA/PCL 70/30 with 3% SiO_2_ displayed an average of 10.61% of Si at the protrusion locations. The results confirmed the affinity of SiO_2_ to be encapsulated by PCL. A better understanding of the interactions between the materials in the presented blends and the quantitative analysis of their morphology could improve the understanding of their properties and allow the optimization of their application for different purposes.

## 1. Introduction

Polymer composites are materials that have been introduced to improve the various functional properties of the material produced. In this context, when bio-based and/or biodegradable materials are used as components of the produced polymer composite, there is an additional environmental benefit, as they promote the circular economy and help to reduce the use of conventional petroleum-based polymers [1,2]. Unfortunately, there are certain limitations to the use of these materials, particularly concerning their inferior properties, which do not match those of conventional materials. In order to improve their performance and utilize their potential, the modification of these materials by blending them with other polymers is the focus of various scientific research.

There are various methods for producing and adjusting the properties of the resulting mixture to the intended application. Some of these are mechanical mixing, i.e., extrusion [3,4,5]; injection molding [6,7]; roller milling, in which the polymers are mixed using heated rollers that can shear and blend the materials together; and solution mixing, in which the polymers are dissolved in a common solvent and then blended together [8,9,10]. It can be used in the pharmaceutical industry, in the plastics industry [11,12], in the textile industry, and elsewhere [13,14,15]. In addition, in situ polymerization is often used to obtain a material with improved properties [11,12], as well as reactive blending, in which chemical reactions between the polymers take place during the mixing process [16,17,18], and other processes in which different mixing methods are combined to obtain acceptable results. 

From an economic point of view, mechanical mixing is most commonly used, as it is the simplest and most cost-effective method in the production of composites. If the components are optimally selected and the mixing conditions carefully defined, the process can result in a homogeneous mixture with improved or customized properties that meet the requirements of the intended application.

One of the most important influences on the properties of the produced material is the degree of miscibility and compatibility of the selected components. Due to the ability to form an improved material, the mixtures obtained can be miscible, partially miscible, and immiscible [19,20]. Miscible polymer composites (single-phase) are homogeneous at the molecular level, optically transparent, and have good mechanical properties. In partially miscible polymer composites, the components of the mixture can be distinguished based on the improved physical and/or chemical properties of the resulting mixture. The immiscible polymer composites produced are heterogeneous, optically opaque, and have poorer mechanical properties than the individual components of the mixture. 

The greatest influence on the properties of a heterogeneous system is the compatibility between the polymer phases in the mixture, which is characterized by the interfacial tension. In a system in which the interactions between the phases are strong, the interfacial tension approaches zero, making the mixture miscible. In systems where the interfacial tensions between the components are high, phase separation occurs, the particle size changes, and certain properties are reduced. This is, actually, the main reason why it is necessary to harmonize and adjust the components with optimal properties in order to prevent phase separation and deterioration of the properties of the mixture obtained. To improve specific properties of polymer composites, various nanofillers and compatibilizers can be added to polymer mixtures, and their application in various polymer blends is constantly increasing [21,22,23]. Currently, very popular nanomaterials used as compatibilizers in two- or three-component matrices are graphene oxide, carbon black, carbon nanotubes, nano-silica, fullerenes, and others [24,25]. Their dispersion, distribution, and interaction within the matrix determine the properties of the resulting composites. They can be used as compatibilizers or nucleating agents to stabilize the morphology of the mixture and improve the mechanical, thermal, and chemical properties of the produced composites. To improve/modify the properties of the produced material, they should reduce the interfacial tension by aligning with the interfaces between the polymer phases.

Poly(lactic acid) (PLA) is considered one of the most interesting biodegradable polymers derived from natural resources, with promising uses in many medical and technical applications, 3D printing, packaging, agriculture, the clothing industry, and other fields [26,27,28,29]. It is the most promising matrix material for the production of sustainable biocomposites, which, with certain adjustments, can overcome the major weaknesses of PLA, such as stiffness and brittleness. To overcome these limitations, PLA is often blended with another polymer or a nanoscale filler to customize the properties of the resulting mixture [30,31,32]. 

Like in other industries, the need to utilize environmentally friendly processes and materials has arisen in the printing industry. Apart from the use of biodegradable materials for packaging production, application in the production of printing plates has become interesting, since printing plates are considered an integral part of the printing process. One of the methods of transferring images from the printing plate to the substrate is embossing. Our previously published study aimed to determine the influence of poly(ε-caprolactone) (PCL) and silica nanoparticles on the mechanical and thermal properties of PLA [33]. 

In that study, the effects of adding PCL and nano-silica in different concentrations to the PLA matrix were investigated. PLA and PCL were chosen due to their biodegradability and the fact that they can be used as an alternative to conventional petroleum-based polymers, which are mainly used for the production of conventional relief printing plates for embossing. The results of the research have shown that a blend of PCL and PLA with the addition of nano-silica has a significant potential to produce materials that match the properties of conventional polymeric materials currently used as relief printing plates and introduce the sustainable approach to using resources in the production of printing plates for embossing. Silica nanoparticles (SiO_2_) were used due to their positive influence on the compatibilization of different polymer materials. 

The research has shown that it was possible to optimize the properties of the observed blends by taking into account their concentration, which must be related to the potential application. It was found that the mechanical properties of the blends were improved by the addition of nanoparticles. In addition, the results showed that the nanoparticles have a positive influence on the thermal stability as well as the thermal degradation of the blends obtained. Furthermore, the results have shown that the addition of smaller amounts of SiO_2_ can contribute to an increase in the storage modulus, indicating a good dispersion and distribution of the nanoparticles in the mixture matrix. 

Despite the positive results in terms of mechanical and thermal properties of the observed PLA/PCL and PLA/PCL/SiO_2_ blends, no quantitative analysis of the morphology and distribution of the fillers in the obtained materials, as well as no analysis of the surface and interfacial properties of the involved components, was performed in that study. In order to obtain detailed information about these properties of the produced composites, this research focuses primarily on the study of the morphology of the PLA-based blends with the addition of PCL and nano-SiO_2_, as well as the distribution and adhesion properties of the involved materials, to contribute to a better understanding of the interactions between the components. 

These interactions and morphology are of high significance for understanding the functionality of the prepared blends in specific applications, since they can affect the thermal, surface, mechanical, and other properties of the material.

## 2. Materials and Methods

### 2.1. Materials

The PLA InegoTM 3251D, Nature Works LLC (Plymouth, MN, USA) was used as a basic matrix. PLA is stiff and brittle below its glass transition temperature (T_g_ = 50–60 °C) and with a melting temperature of T_m_ = 170 °C [34]. The tensile strength of neat PLA is 60 MPa, and tensile elongation equals 3.5%. PCL Capa 6800, Perstorp (Warrington, UK) was added to PLA in a certain weight ratio. The glass transition and the melting temperatures of PCL are −60 °C and 60 °C, respectively [35]. The tensile strength of neat PCL is 20 MPa, and tensile elongation equals 800%. 

The first set of samples was prepared by adding a PCL in the PLA matrix, up to 50 wt.% of PLA. The following blend compositions of PLA/PCL were obtained: PLA/PCL 100/0, 90/10, 80/20, 70/30, 60/50, and 50/50. The fumed silica (Aerosil^®^200) was kindly supplied by Evonik (Hanau, Germany), CAS No. 112945–52–5. The average particle size of the silica was 12 nm, and it was used as received without any pre-treatment. In order to observe the interaction of PLA and PLC with nano-silica, silica was added by mixing a new set of samples with the same concentration of PLA and PCL and with added silica at concentrations of 1 wt.% and 3 wt.%. The following blend compositions of PLA/PCL/SiO_2_ were obtained: PLA/PCL/SiO_2_ 100/0/1, 100/0/3, 90/10/1, 90/10/3, 80/20/1, 80/20/3, …, 50/50/1, and 50/50/3.

To form a PLA-based polymer blend, the materials were blended in the Brabender^®^ (Duisburg, Germany) internal mixer for 5 min at a temperature of 190 °C. After the formation of a homogeneous mixture, it was cut to pieces and molded into plates with dimensions of 100 mm × 100 mm × 1.4 mm. The hydraulic press was used for 7 min (2 min of preheating and 5 min of pressing) at a temperature of 190 °C and a pressure of 16 MPa production of plates. Overall, one set of six plates was prepared with PLA/PCL components and another set of twelve samples with PLA/PCL/SiO_2_ components, which were ready for analysis.

### 2.2. Characterization Methods

#### 2.2.1. Surface Morphology and Cross-Sections of Samples

In order to observe the surface morphology and cross-sections of the produced materials, a scanning electron microscope (SEM) was used (JSM–6060LV, Jeol, Tokyo, Japan). Before the imaging, the high vacuum evaporation process was used for coating samples with a thin gold layer.

#### 2.2.2. Contact Angles, Surface Free Energy, and Adhesion Parameters

Contact angles of the referent liquids on the polymer blends were measured using a DataPhysics OCA 30 goniometer (DataPhysics Instruments GmbH, Filderstadt, Germany). Contact angles were measured using the sessile drop method. Ten drops with a volume of 1 µL of each referent liquid with known surface tension were applied to the samples at different spots. Demineralized water, diiodomethane, and glycerol were used as referent liquids. Total, dispersive, and polar surface tension components of probe liquids expressed in mJ/m^2^ were: diiodomethane—50.8, 50.8, and 0; glycerol—64.0, 34.0, and 30.0; and water—72.8, 21.8, and 51.0; respectively. The measurements of the contact angle were performed at the same moment after the droplet touched the sample surface, and the average value of 10 measurements was calculated. The contact angle of water on the polymer blends was of special interest for providing basic information on the changes in hydrophilicity of the surface with the changed ratios of the components in PLA/PCL/SiO_2_ blends.

After the average value of the contact angle was calculated for each of the three liquids on each polymer blend, these values were used to determine the surface free energy (SFE) and its polar and dispersive component, using the OWRK method [36]. The determination of SFE and its components was carried out to gain insight into the surface properties of the PLA/PCL/SiO_2_ blends and to determine the adhesion parameters between the components in the blends. The obtained results contributed to a detailed morphological and surface characterization of the blends.

Adhesion parameters between the components in the PLA/PCL/SiO_2_ blends can be described and expressed by observing the two-component and three-component systems. Adhesion parameters of two-component systems are surface free energy of the interphase (γ_12_), thermodynamic work of adhesion (*W*_12_), and spreading coefficient (*S*_12_). The thermodynamic work of adhesion (*W*_12_) (Equation (1)) is defined as the work required to overcome the forces of attraction between two different molecules in a liquid or solid. It is used for the prediction of interactions at the interface of two phases in contact. A higher amount of work of adhesion signifies better adhesion [37,38,39].
(1)$$W_{12}=\gamma_{1}+\gamma_{2}-\gamma_{12}$$

The subscripts refer to the SFE of the solids in contact, and the γ_12_ denotes the surface free energy of their interphase. The surface free energy of the interphase was calculated according to Equation (2):(2)$$\gamma_{12}=\gamma_{1}+\gamma_{2}-2\;\sqrt{\gamma_{1}^{d}\;}\gamma_{2}^{d}-2\;\sqrt{\gamma_{1}^{p}\;}\gamma_{2}^{p}$$

Furthermore, the positive spreading coefficient (*S*_12_, Equation (3)) indicates a spontaneous wetting:(3)$$S_{12}=\gamma_{1}-\gamma_{2}-\gamma_{12}$$
where γ_1_ is the SFE between the first solid surface and vapor, γ_2_ is the SFE between the other solid surface and vapor, and γ_12_ is the SFE between the solid surfaces in contact. 

In the three-component system, there are potentially three different arrangements of dispersed phases in the matrix. If X is a matrix, and Y and Z are dispersed phases, they can remain separated in the X matrix. Furthermore, the Y phase can encapsulate the Z phase, or the Z phase can encapsulate the Y phase. In a three-component system, there is only one parameter that describes the adhesion, and that is the spreading coefficient (*S*_23_ or *S*_32_). The coefficient *S*_23_ represents the tendency of the dispersed phase Y(_2_) to encapsulate the dispersed phase Z(_3_) within the matrix X(_1_):(4)$$S_{23}=\gamma_{31}-\gamma_{21}-\gamma_{23}$$

The coefficient *S*_32_ represents the tendency of the dispersed phase Z(_3_) to encapsulate the dispersed phase Y(_2_) within the matrix X(_1_):(5)$$S_{32}=\gamma_{21}-\gamma_{31}-\gamma_{23}$$

Encapsulation will occur if the parameter *S* has a positive value. The parameters *S*_23_ and *S*_32_ are also called Harkin coefficients, and they give a better insight into the morphology of the mixture [40]. In Equations (4) and (5), the parameter γ denotes the interfacial tension between different pairs of matrix and dispersed phases.

In this work, the adhesion parameters were calculated to determine whether the mixing of two otherwise immiscible polymers took place and whether the addition of SiO_2_ nanoparticles could contribute to the improvement of miscibility. In the three-component system, PLA represents the matrix, and PCL and SiO_2_ nanoparticles are the dispersed phase. The case where PCL was the matrix and PLA the dispersed phase was analyzed as well because of the varied ratio of PLA to PCL up to 1:1. 

If the coefficient *S*_23_ takes on a positive value, it means that SiO_2_ is located in the dispersed polymer (PCL), that is, that SiO_2_ is encapsulated by the dispersed polymer. Suppose the coefficient *S*_23_ takes on a negative value. In that case, it means that the SiO_2_ nanoparticles have settled into the matrix, that is, that SiO_2_ and the dispersed polymer have remained separated phases within the matrix. The third case is when the value of the coefficient *S*_23_ would be close to zero, in which case SiO_2_ would be placed at the polymer interface, which would contribute to the reduction in the interfacial energy, resulting in a finer morphology of the blend.

#### 2.2.3. FTIR-ATR Analysis of PLA/PCL and PLA/PCL/SiO_2_ Blends 

Fourier transform infrared spectroscopy–attenuated total reflectance (FTIR-ATR) was performed to provide information about the chemical nature and molecular structure of the samples, as well as to analyze the differences in the sample structure related to the PLA/PCL ratio and the amount of added SiO_2_ nanoparticles. FTIR-ATR analysis was performed using the IRAffinity–1 FTIR Spectrophotometer (Shimadzu, Kyoto, Japan). The crystal type was ZnSe (index of refraction 2.4), number of scans was 15, and the resolution was 4 cm^−1^. IR spectra were recorded in the spectral range between 3000 and 600 cm^−1^, and the areas of interest (with detectable changes) were displayed.

#### 2.2.4. SEM-EDS Analysis

Surface morphology and chemical composition analyses of the samples were conducted using the ThermoFischer Scientific Quattro S Field-Emission Scanning Electron Microscope (FEG-SEM) (Thermo Fisher Scientific Inc., Waltham, MA, USA), equipped with an Oxford Instruments 64 Energy-Dispersive Spectrometer (EDS) (Abingdon, UK). The instrument operated in low-vacuum mode, with an acceleration voltage of 10 kV and a beam current of 10 nA. EDS analyses were performed at a working distance of 10 mm, and spectrum acquisition occurred over a 60 s period. 

The samples were firstly cut at 30–45° with an IsoMet low-speed saw (Buehler, IL, USA) with sintered cubic boron nitride (CBN) cutting blades. The cut surfaces were then sanded first with 60, 80, 120, 600, 800, and 1200 grit paper and then polished with 3M polyester polish. Finally, the samples were cleaned with soap in an ultrasonic bath for 5 min to remove residual hydrocarbons. 

The samples were mounted onto an aluminum sample holder in such a way that the cut section of each sample was perpendicular to the direction of the electron beam. Along the edges, carbon tape was attached to enable the flow of electrons of the sample and prevent charging. 

## 3. Results

### 3.1. Morphology and Cross-Section of Samples

The morphology of PLA/PCL and PLA/PCL/SiO_2_ blends was used to observe the possible distribution and interfacial bonding between the materials. SEM images of these samples are shown in Figure 1a–c. 

In our previous research [33], we found that PLA/PCL samples without nanoparticles have two distinct phases in the surface fracture, with spherical domains of dispersed PCL in PLA. In Figure 1a, SEM images show the PLA matrix and the dispersed PCL domains, taken at 3000× magnification. The spherical PCL domains are marked with black arrows and correspond to the typical morphology of sea-island morphology [30]. The sample with the same amount of components, the PLA/PCL 50/50 sample, shows visible cracks and slight delamination of the materials (marked with white arrows), indicating that PLA and PCL are incompatible and immiscible polymers. 

In the observed PLA/PCL samples with 1% SiO_2_, spherical PCL domains are still visible, especially in the PLA/PCL/SiO_2_ 90/10/1 and 80/20/1 samples. In the samples with a higher PCL content, the spherical elements have been reduced, indicating better interactions at the interface between PLA, PCL, and 1% SiO_2_. On the samples, tiny elements, probably agglomerates of nano-SiO_2_, are visible in the images (marked with red arrows), indicating the possible placement of SiO_2_ nanoparticles in the PCL domain, which was confirmed by the results presented in the continuation of this research. There were no visible changes on the PLA/PCL images with a higher amount of SiO_2_ nanoparticles in comparison to 1% SiO_2_.

### 3.2. Surface Properties and Adhesion Parameters of PLA/PCL and PLA/PCL/SiO_2_ Blends

The contact angle of water on the solid is indicative of the changes in the hydrophilicity of a solid material. Measured contact angles of water on PLA/PCL/SiO_2_ blends are presented in Figure 2. 

From the results of the water contact angle on polymer blends, it is evident that the blends with 0% of SiO_2_ nanoparticles and with PCL in the samples display a lower water contact angle than pure PLA. The reason for the decreased contact angle of water when PCL is added to PLA is the higher polar component of PCL’s SFE (Table 1). Furthermore, measured contact angles of water indicate that the addition of SiO_2_ nanoparticles leads to an increase in the contact angle, i.e., the surface becomes more hydrophobic. Such results are contrary to initial expectations because, considering that SiO_2_ nanoparticles (Aerosil 200) are hydrophilic, the contact angle of water on the samples should decrease.

The reason for such results may be in the surface of SiO_2_ nanoparticles, which contain hydrophilic silanol areas (Si–OH) and hydrophobic areas of siloxane bridges (Si–O–Si) that can be activated at elevated temperatures. The formation of hydrophobic areas can occur at a temperature of 177 °C and above (the temperature during the preparation of blends was 190 °C) when the loss of hydroxyl pairs begins, and the silica surface slowly becomes hydrophobic. Hydrophobicity can also occur because of silanol condensation in siloxane bridges. Ultimately, due to the decrease in the number of silanols and the increase in the number of siloxane bridges on the surface, there is an increase in the value of the contact angle of water on samples containing SiO_2_ nanoparticles [41,42,43,44]. 

Table 1 and Figure 3 show the results of surface free energy (SFE) values of the components in polymer blends and SFE of the blends, respectively. Surface free energy was determined by the OWRK method, and the table shows the values of the dispersive, polar, and total SFE. SFE components of pure PLA, PCL, and SiO_2_ were needed for the calculation of the adhesion parameters between the components in the PLA/PCL/SiO_2_ system.

Components of the SFE of pure PLA and PCL were calculated from the values of referent liquids’ contact angles, and the SFE of SiO_2_ nanoparticles (Aerosil 200) was taken from the literature [45].

The obtained results of PLA/PCL/SiO_2_ blends’ SFE indicate that the addition of nanoparticles caused a decrease in the SFE. The reason for this occurrence could be the mentioned surface properties of the nanoparticles. The highest total SFE (43.61 mJ/m^2^) was calculated on the sample without the addition of nanoparticles, specifically, the PLA/PCL/SiO_2_ blend 60/40/0 (Figure 3a). The lowest total SFE values were generally calculated on the blends with 1% SiO_2_ (specifically, 34.83 mJ/m^2^ on the blend 60/40/1, Figure 3b). Blends with 3% SiO_2_ probably had higher SFE values than blends with 1% SiO_2_ (Figure 3c) because of the SiO_2_ agglomeration detected in EDS mapping. Moreover, the polar component of the SFE was low (below 10 mJ/m^2^) for all samples and decreased further after the addition of the nanoparticles to the blends. The change in the proportion of PCL in the mixture did not significantly affect the change in the total surface free energy of the samples, since the difference in the SFE values of PLA and PCL are not that significant.

Table 2 shows the calculated adhesion parameters in two-component and three-component systems. The optimal conditions for achieving adhesion in a two-component system are the maximum thermodynamic work of adhesion ($W_{12}$), a positive value of the spreading coefficient ($S_{12}$), and a minimum value of the free interfacial energy ($\gamma_{12}$). In the case of a three-component system, if the coefficient $S_{23}$ takes on a positive value, it means that the SiO_2_ compatibilizer is encapsulated by the dispersed polymer (PCL). If $S_{23\;}$ has a negative value, it means that the SiO_2_ nanoparticles have settled in the matrix (PLA), that is, that the compatibilizer and the dispersed polymer have remained In separate phases within the matrix. The third case would be if the value of $S_{23}$ is close to zero, in which case the nanoparticles are placed at the PLA/PCL interface, thereby contributing to the reduction in the energy of the interface between the two polymers.

Of the three analyzed two-component systems (PLA/PCL, PLA/SiO_2_, and PCL/SiO_2_), the optimal adhesion parameters were achieved for the PLA/PCL system. Although the other two systems have a slightly higher thermodynamic work of adhesion, the PLA/PCL system has the lowest value of the interfacial tension (1.79 mJ/m^2^) and a positive value of the spreading coefficient (4.10 mJ/m^2^).

In the three-component system PLA/PCL/SiO_2_, $S_{23}$ was 9.51 mJ/m^2^, which means that in such a system, the nanoparticles should be located within the dispersed (PCL) phase. Since the ratio of PLA to PCL in this research reached 1:1, the case of $S_{23}$ if PLA was a dispersed phase in the PCL matrix was calculated, and a negative value of −13.09 mJ/m^2^ was obtained, confirming the placement of SiO_2_ nanoparticles in PCL.

In addition to the adhesion parameters, the location of the nanoparticles within a system is also influenced by the polymer viscosity [46]. The lower viscosity of a polymer enables easier encapsulation of nanoparticles by that polymer. PCL has a lower viscosity and higher polar part of the surface free energy compared to PLA, and this certainly influenced the placement of nanoparticles within PCL. Also, the melting temperature plays an important role because a polymer that melts at a lower temperature encapsulates nanoparticles more easily than one that melts at a higher temperature. This feature is also in favor of PCL because it melts at a lower temperature than PLA [47].

### 3.3. FTIR-ATR Spectra of PLA/PCL and PLA/PCL/SiO_2_ Blends

FTIR-ATR spectra of all samples can be observed in Figure 4. The specific areas of functional groups for PLA and PCL in general are highlighted.

At 2993 cm^−1^, a band belonging to C–H asymmetric stretching in CH_3_ appears in PLA [41,42,43]. With an increase in the proportion of PLA in the mixture, this band moves to higher values (from 2995 to 2997 cm^−1^). The next band belongs to the symmetric stretching of C–H in CH_3_, which is located at a lower wavelength than that of the asymmetric stretching (from 2926 to 2943 cm^−1^) and is specific for PLA [48,49,50]. Also, the asymmetric stretching of C–H in CH_2_, which occurs in PCL, is specific for this interval. Symmetric stretching of C–H in CH_2_ of PCL occurs at 2862 cm^−1^ [51,52]. With an increase in the proportion of PLA in the mixture, it moves towards lower values, so in samples with pure PLA, this band appeared at 2854 cm^−1^. The next visible band represents the stretching of the C=O bond, which is characteristic of polyester materials such as PLA and PCL [50]. It appears in the interval from 1725 to 1745 cm^−1^. Asymmetric and symmetric stretches of the CH_3_ group, specific for PLA, appear at 1452 and 1357 cm^−1^ for pure PLA, and with the addition of PCL in the mixture, these values shift to 1456 and 1363 cm^−1^ for samples with 50% of the proportion of PCL in the mixture. C–O–C asymmetric and symmetric stretches in PLA are visible at values around 1180 and 1080 cm^−1^ [48,49,50]. Furthermore, in samples PLA/PCL/SiO_2_ 50/50/0, PLA/PCL/SiO_2_ 50/50/1, and PLA/PCL/SiO_2_ 50/50/3, a band at 1236 cm^−1^ corresponding to C–O–C asymmetric stretching of PCL can be observed [53]. CH rocking occurs at 1126 cm^−1^ for pure PLA, and with the addition of PCL, this band moves to higher values—in samples with 50% of PCL in the mixture, it is at 1136 cm^−1^ [49,50]. In the Interval from 1035 to 1039 cm^−1^, there is a band corresponding to C–CH_3_ stretching. The amorphous and crystalline phases appear In the region below 1000 cm^−1^ [49].

Figure 5 presents more detailed, specific changes in chosen FTIR – ATR spectra of PLA/PCL/SiO_2_ blends with different concentrations of the nanoparticles.

In Figure 5a, FTIR – ATR spectra of PLA/PCL 100/0 blends are presented. It is noteworthy that the band at 1645 cm^−1^ of low intensity, belonging to the bound water, is observable regardless of the SiO_2_ concentration. This can be related both to the hygroscopicity of pure PLA and to the agglomeration of the nanoparticles that occurs at their higher concentration. This occurrence is more expressed in pure PLA than in the PLA/PCL blend, according to the EDS mapping. 

Figure 5b presents FTIR – ATR spectra of PLA/PCL 70/30 blends. The areas of interest are similar to the areas for the 50/50 blend (Figure 5c). However, the band at 1267 cm^−1^ is pronounced only for the 70/30/3 blend. This band is associated with C=O bending vibration [54]. It was generally observed in spectra with 3% of nanoparticles and on the spectra of the blends 100/0 and 50/50 (Figure 4). This points to the changes in the interactions of PLA and PCL in the polymer blends, related to the PLA/PCL ratio. 

In Figure 5c, FTIR – ATR spectra of PLA/PCL 50/50 blends are presented. Characteristic bands related to the crystalline phase of PLA can be observed around 955 and 922 cm^−1^. The amorphous phase can be observed at 908 cm^−1^ [55,56,57]. Furthermore, the band at 1135 cm^−1^, which is also indicative of the crystalline phase [58], shifted to 1130 cm^−1^ [59] for the 50/50/3 blend. This indicates that nanoparticles in the PLA/PCL 50/50 blend influence the structure ordering of PLA in the blend, which is aligned with the results obtained in the previous research [33]. The low-intensity band around 1645 cm^−1^ is interesting since it represents water absorbed by hygroscopic PLA [60,61]. The work from other authors [62] suggests that both hydrophilic and hydrophobic nano-silica can be used to reduce the hygroscopicity of PLA. According to FTIR – ATR spectra in Figure 5a–c, Aerosil 200 showed only a mild effect on the reduction in the bound water in blend 50/50/3. 

Finally, the bands at 1725 and 1747 cm^−1^ represent the vibration of the carbonyl bond. Specifically, the band at 1725 cm^−1^ represents C=O vibration in crystalline PCL, and the band at 1747 cm^−1^ represents C=O vibration in the ester group of PLA [63,64]. 

### 3.4. Microstructure and Elemental Distribution (SEM-EDS Analysis)

The surface of the PLA/PCL/SiO_2_ 100/0/1 sample was initially without pronounced textures or protrusions. Short-term (minutes) exposure to the electron beam etched the surface, revealing thin Si- and O-rich linear structures that protruded from the surface of the sample, giving the sample a distinct topographical dimension. Moreover, in the sample PLA/PCL/SiO_2_ 100/0/1, a linear elemental analysis (SEM-EDS) was performed. The analysis site was selected to cross both the matrix and the silica-rich linear domain, as shown in Figure 6. In the profile, the light-colored graph represents the raw signal intensity line, whereas the dark-colored graph denotes the averaged intensity line, providing a clearer interpretation of elemental distribution. The elemental distribution along the line-scan profile reveals a notable increase in the concentrations of Si and O at the locations of the protruding lines. At the same time, a decrease in the concentration of C is observed, indicating the presence of a Si–O phase.

The results of EDS point analysis are summarized in Table 3. The measured Si content in the analyzed material is in a range between 0.50 and 1.44 wt.%, with an average of 1.05 wt.%, which closely corresponds to the quantity of SiO_2_ added to the sample. 

In the sample PLA/PCL/SiO_2_ 100/0/3, SEM-EDS point elemental analysis (Table 4) and mapping were conducted (Figure 7). The EDS element distribution map reveals homogenously distributed Si-rich particles (depicted red), which appear densified in certain regions of the sample. Consequently, separate analyses were performed for the PLA matrix and the Si-rich densifications. The red areas observed on the sample surface suggest the potential presence of SiO_2_ clusters (Figure 7).

The results of SEM-EDS point analysis of the sample PLA/PCL/SiO_2_ 100/0/30 are given in Table 4. The average Si content within the PLA matrix is 2.84 wt.%, aligning well with the quantity of SiO_2_ added to the sample. In contrast, the results of analysis of the densifications (red dots, Figure 7) reveal that the Si content is more than ten times higher (31.54 wt.%), suggesting the formation of SiO_2_ clusters or segregations due to elevated SiO_2_ concentrations in the PLA matrix.

In the SEM images of sample PLCA/PCL/SiO_2_ 90/10/1 (Figure 8), it is evident that thin linear structures protruding from the sample surface, similar to those observed in sample PLA/PCL/SiO_2_ 100/0/1, remain visible at high magnifications. However, with the addition of 3% SiO_2_ to the PLA/PCL 90/10 mixture, the structure adopts a more granular appearance.

The EDS point analysis results of the sample PLA/PCL/SiO_2_ 90/10/1 presented in Table 5 indicate a notable difference in Si concentrations between areas without protrusion (0.58 wt.%) and those with protrusions (2.76 wt.%). Similarly, in sample PLA/PCL/SiO_2_ 90/10/3, a comparable trend is observed; however, with a less pronounced difference in Si concentration compared to the sample with 1% SiO_2_ addition. Additionally, in sample PLA/PCL/SiO_2_ 90/10/3, the distribution of Si in areas without protrusions may be uneven, as evidenced by a significant difference between the minimum and maximum values of Si content.

The EDS analysis was conducted on both the PLA and PCL components in sample PLA/PCL/SiO_2_ 70/30/1, as shown in Figure 9. Protrusions on the surface, potentially SiO_2_ clusters, are represented by “Light”, while the PLA matrix is indicated in “Grey” and the PCL component in “Black”. In the sample PLA/PC/SiO_2_ 70/30/3, analysis was limited to two different areas: the PLA matrix (denoted by “Light”) and the PCL component (denoted by “Black”). 

The EDS analysis results for both samples are summarized In Table 6.

Table 6 reveals that the silicon (Si) concentration in the PCL component is consistently higher in both samples compared to the PLA matrix. This outcome is to some extent expected, as the glass transition and melting point of PCL is at a lower temperature than that of PLA, and thus, the viscosity during mixing is higher, resulting in easier mixing with SiO_2_ and taking up its higher fraction. Moreover, the interfacial properties described in Table 2 support SiO_2_ dispersion within PCL. Furthermore, the interfacial properties of the components in the polymer blend (Table 2) are in accordance with SiO_2_ having an affinity of being dispersed in PCL. However, areas with higher Si concentrations (protrusions) are still present in the sample mixture (probably clusters of SiO_2_).

Figure 10 presents the surface morphology of PLA/PCL 50/50 samples with the addition of nano-SiO_2_. In both samples with 1% and 3% SiO_2_, lighter and darker areas represent the PLA matrix and PCL added component, respectively.

SEM-EDS linear elemental analysis was performed on the PLA/PCL/SiO_2_ 50/50/3 sample. As depicted in Figure 11, the PLA matrix exhibits a lower Si content compared to the PCL areas, where the Si content is either close to or below the limit of quantification. This finding correlates with the observations derived from the calculated spreading coefficients in a three-component system (refer to Table 2).

As indicated in Table 7, the concentrations of carbon (C) and oxygen (O) are comparable in both areas, while the difference lies in the silicon (Si) content. It is expected that the PLA component would have a lower percentage of Si compared to PCL. Notably, both components exhibit a significant variation in Si content (as indicated by the range of minimum and maximum percentages), suggesting the possibility of uneven distribution of SiO_2_ within both components. 

The results of this research are significant due to the observed impact of morphology and interfacial properties on other properties of the PLA/PCL/SiO_2_ blends. Past research [26] has reported that the addition of nano-SiO_2_ to PLA/PCL blends with higher portions of PCL (40–50%) can increase tensile strength and strain at break. This appears to be due to a reduction in the size of PCL domains, which was confirmed in this study. The addition of SiO_2_ in the correct concentration can improve the mechanical properties of the blends, especially stiffness and toughness, while contributing to an increase in the storage modulus through good dispersion and distribution within the matrix. However, an increase in the concentration of SiO_2_ can decrease the strengthening effect on the material due to nanoparticle agglomeration, which causes poor dispersion and distribution within the mixture, resulting in surface irregularities and the creation of voids within the material structure. The addition of SiO_2_ in the PLA/PCL blends also increased the maximum temperature degradation rate, which improved the thermal degradation behavior of observed blends, especially for blends modified with 3% SiO_2_. This improvement in thermal stability could be attributed to the interaction between well-dispersed nanoparticles and the polymer matrix.

## 4. Conclusions

The objective of this study was to improve the understanding of how PLA, PCL, and hydrophilic nano-SiO_2_ interact in a blend that can be used in various industries and other applications (for example, as a material for a relief printing plate made for the embossing process). Eighteen samples of PLA/PCL/SiO_2_ blends were produced by varying the concentration of all components. SEM analysis, EDS mapping, water contact angle measurements, surface free energy calculation, and adhesion parameter measurements were conducted on the samples. FTIR-ATR analysis was also performed to determine the changes in the characteristic functional groups of the polymers used.

Results have shown that the addition of SiO_2_ nanoparticles leads to an increase in the contact angle of water, i.e., the surface becomes more hydrophobic. The reason for such results may be hydrophilic silanol areas (Si–OH) and hydrophobic areas of siloxane bridges (Si–O–Si) that can be activated at elevated temperatures when the loss of hydroxyl pairs can begin, thereby affecting the surface properties of the material. Furthermore, in the three-component systems, calculated spreading coefficients indicated that SiO_2_ nanoparticles have the affinity of being placed within the dispersed (PCL) phase. SEM images of the blends with all three components showed the spherical PCL corresponding to the typical morphology of the sea island. In the samples with a higher PCL content, the spherical elements have been reduced, indicating better interactions at the interface between PLA, PCL, and SiO_2_.

FTIR-ATR analysis showed that hydrophilic SiO_2_ nanoparticles showed a mild effect on the reduction in the water bound by hygroscopic PLA. Shifts in the bands representing the crystalline phase of PLA indicated that nanoparticles influence the structure ordering of PLA in the blend with equal portions of PLA and PCL. 

SEM-EDS mapping of the PLA/PCL 100/0 blend surfaces revealed the presence of SiO_2_ clusters, and silicon (Si) concentration reaching up to ten times higher than the nominal concentration of SiO_2_. However, with the addition of 3% SiO_2_ to the blend containing PCL, the structure became more granular.

A better understanding of the morphology and interactions between materials in the presented blends, as well as the advantages and disadvantages of their varied composition, can lead to optimization of their application for different purposes.

## Figures and Tables

**Figure 1 polymers-16-01739-f001:**
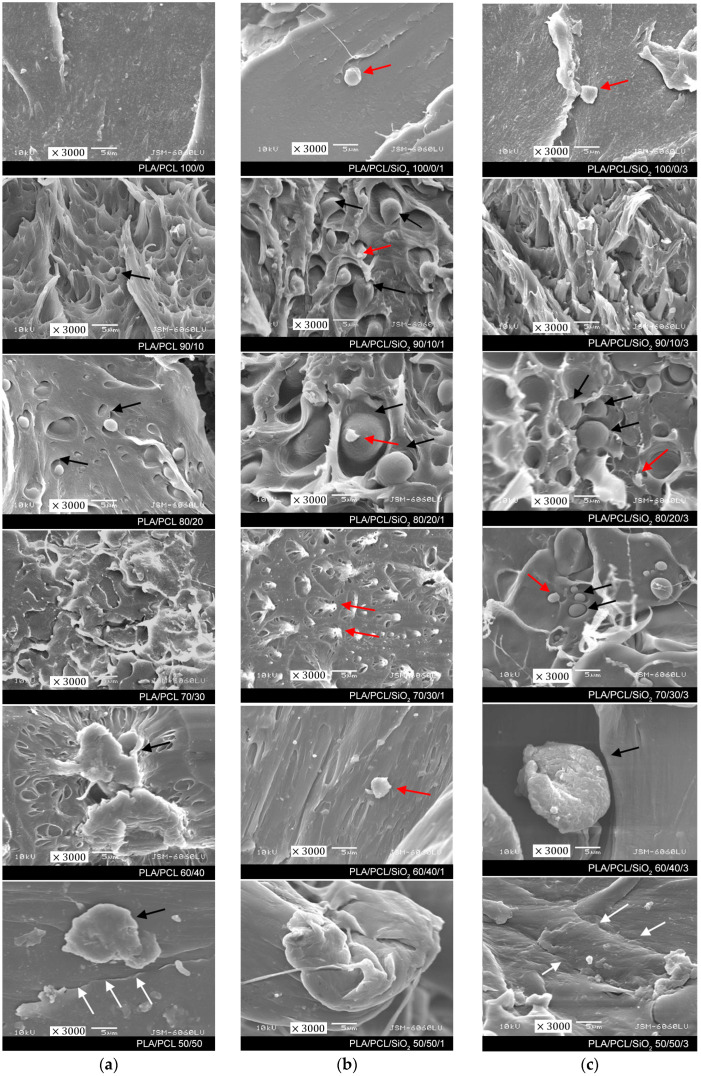
SEM micrographs of fracture surfaces of PLA/PCL/SiO_2_ samples: (**a**) without SiO_2_, (**b**) with 1% SiO_2_, and (**c**) with 3% SiO_2_ (mag. 3000×). Black arrows show spherical PCL domains, red arrows show agglomerates of nano-SiO_2_, and white arrows mark the cracks and slight delamination of the materials.

**Figure 2 polymers-16-01739-f002:**
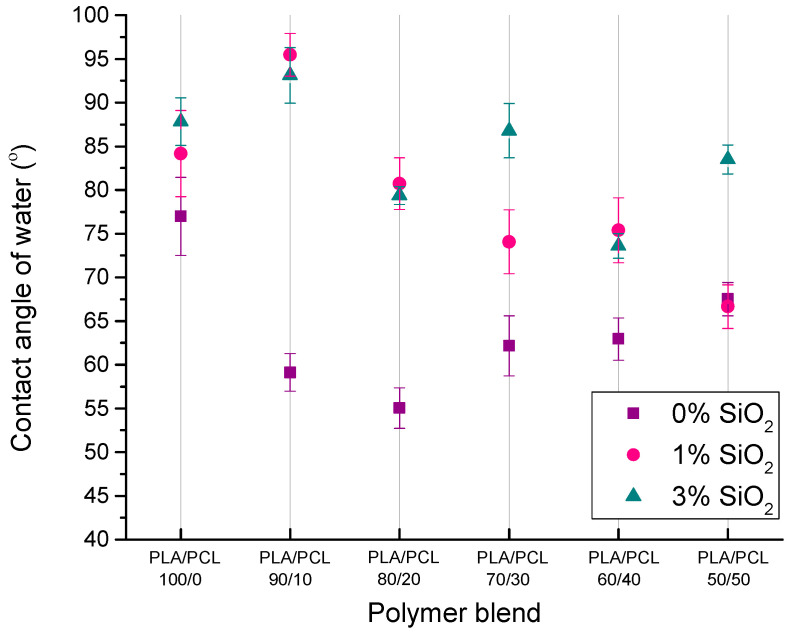
Contact angle of water on PLA/PCL and PLA/PCL/SiO_2_ blends.

**Figure 3 polymers-16-01739-f003:**
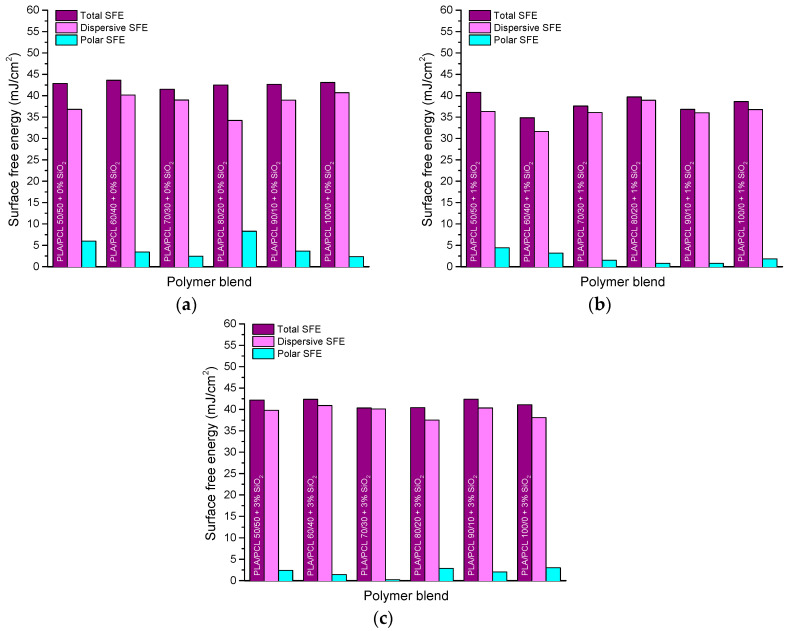
Surface free energy of PLA/PCL blends with: (**a**) 0% SiO_2_, (**b**) 1% SiO_2_, and (**c**) 3% SiO_2_.

**Figure 4 polymers-16-01739-f004:**
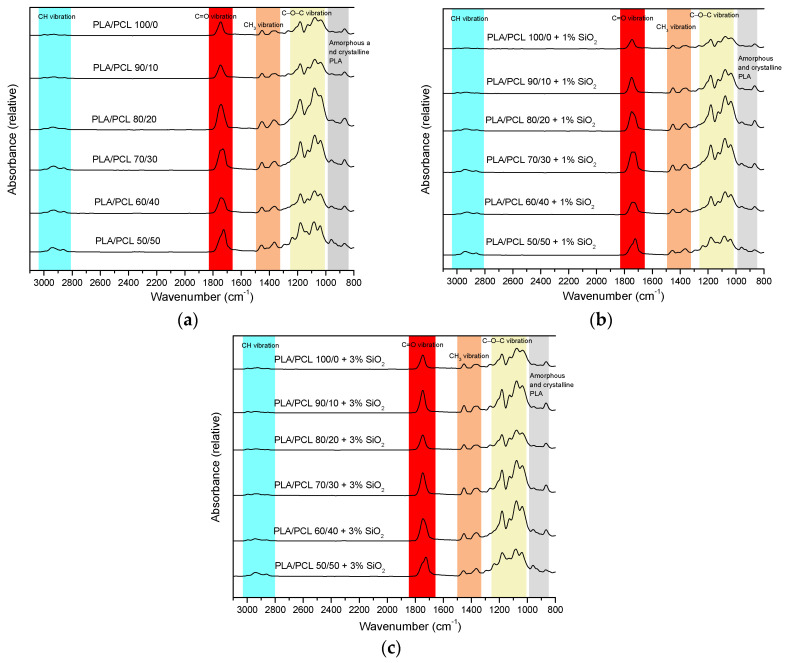
FTIR – ATR spectra of PLA/PCL blends with: (**a**) 0% SiO_2_, (**b**) 1% SiO_2_, and (**c**) 3% SiO_2_.

**Figure 5 polymers-16-01739-f005:**
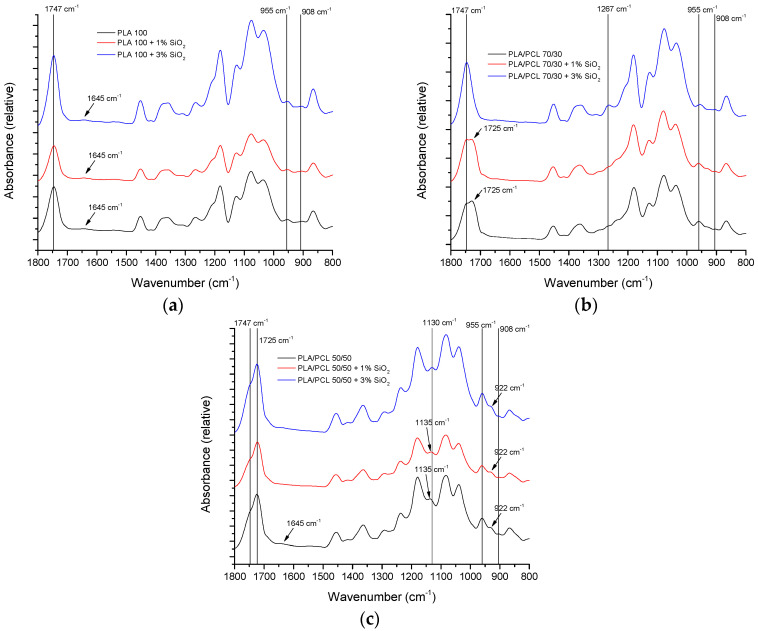
FTIR – ATR spectra of PLA/PCL blends: (**a**) PLA/PCL 100/0, (**b**) PLA/PCL 70/30, and (**c**) PLA/PCL 50/50.

**Figure 6 polymers-16-01739-f006:**
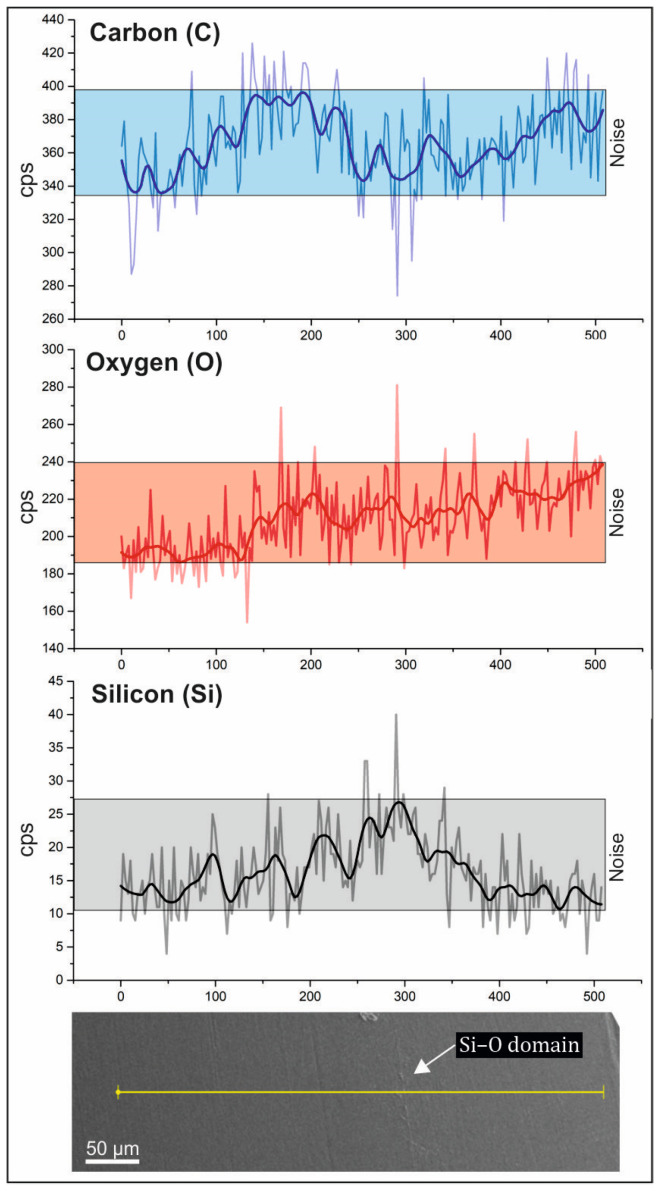
The SEM – EDS line-scan profile of sample PLA/PLC/SiO_2_ 100/0/1 (light – colored graph represents the raw signal intensity line, whereas the dark – colored graph denotes the averaged intensity line).

**Figure 7 polymers-16-01739-f007:**
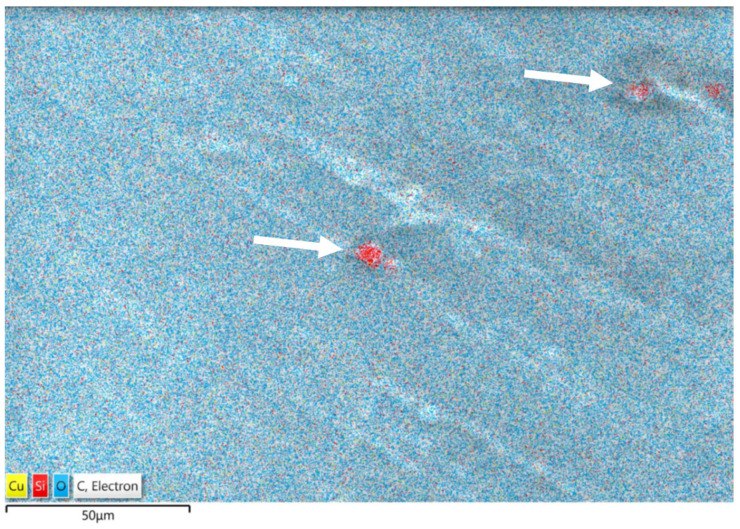
The EDS element distribution map of sample PLA/PCL/SiO_2_ 100/0/3 with the Si-rich particles (arrows indicate Si-rich densifications).

**Figure 8 polymers-16-01739-f008:**
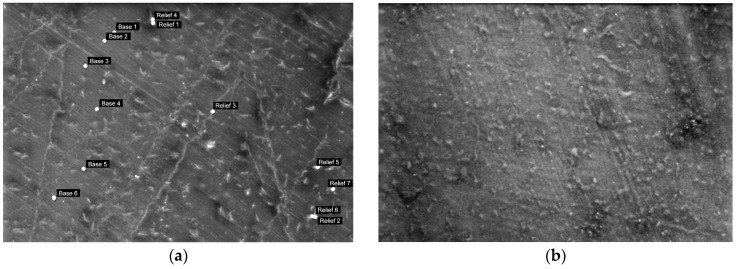
Surface morphology of the sample PLA/PCL 90/10 with added (**a**) 1% SiO_2_ and (**b**) 3% SiO_2_ (mag. 1000×).

**Figure 9 polymers-16-01739-f009:**
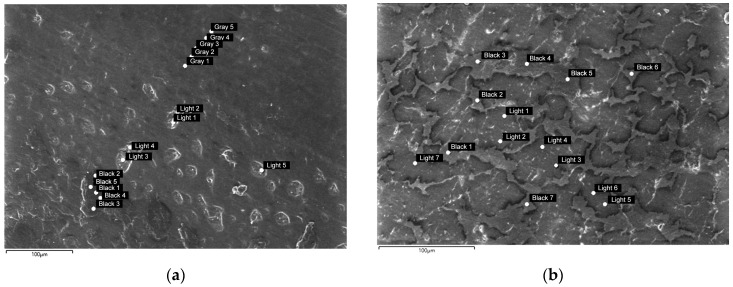
Surface morphology of the sample PLA/PCL 70/30 with added (**a**) 1% SiO_2_ and (**b**) 3% SiO_2_. The points on the images indicate areas analyzed by the SEM-EDS.

**Figure 10 polymers-16-01739-f010:**
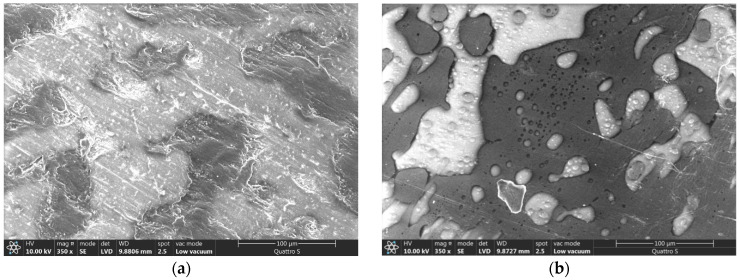
PLA matrix (lighter areas) and PCL component (darker areas) in the sample PLA/PCL 50/50 with added (**a**) 1% SiO_2_ and (**b**) 3% SiO_2_ (mag. 350×).

**Figure 11 polymers-16-01739-f011:**
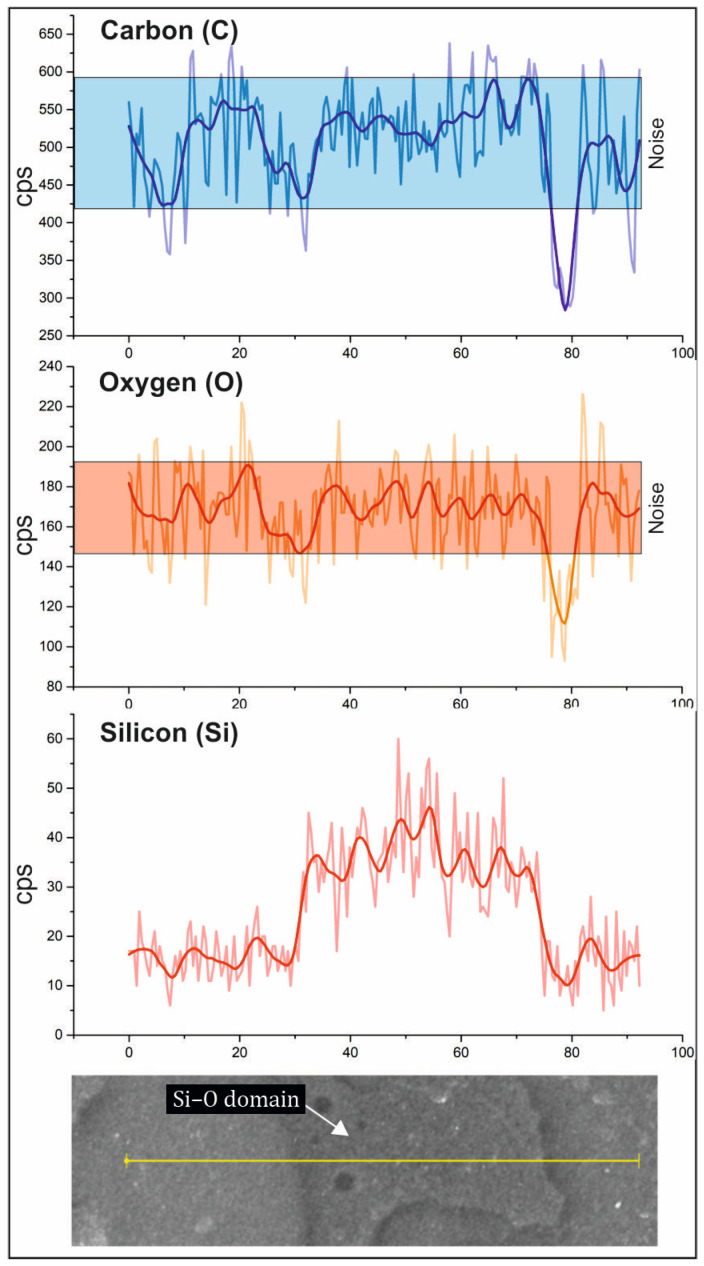
The line − scan profile of the PLA/PCL/SiO_2_ 50/50/3 sample (light − colored graph represents the raw signal intensity line, whereas the dark − colored graph denotes the averaged intensity line).

**Table 1 polymers-16-01739-t001:** Surface free energy components of pure PLA, pure PCL, and nano-SiO_2_ (Aerosil 200).

Component	γ ^dispersive^	γ ^polar^	γ ^total^
PLA	40.72	2.38	43.10
PCL	30.59	6.61	37.21
SiO_2_—Aerosil 200 [6]	29.40	50.60	80.00

**Table 2 polymers-16-01739-t002:** Adhesion parameters in two-component and three-component systems.

System	Two-Component System			Three-Component System
	$\gamma_{12}$	$W_{12}$	$S_{12}$	*S* _23_
PLA/PCL	1.79	78.52	4.10	/
PLA/SiO_2_	31.95	91.15	−68.85	/
PCL/SiO_2_	20.65	96.56	−63.44	/
PLA/PCL/SiO_2_	/	/	/	9.51
PCL/PLA/SiO_2_	/	/	/	−13.09

**Table 3 polymers-16-01739-t003:** Content of C, O, and Si in the sample PLA/PCL/SiO_2_ 100/0/1 (all data given in wt.%).

Element	C [%]	O [%]	Si [%]
**Min. **	58.93	32.01	0.50
**Max.**	66.95	39.92	1.44
**Average**	66.14	37.55	1.05
**S.D.**	3.04	2.99	0.26

**Table 4 polymers-16-01739-t004:** Content of C, O, and Si in the sample PLA/PCL/SiO_2_ 100/0/3.

Sample	PLA Matrix			Densifications (SiO_2_ Clusters)		
Element	C [%]	O [%]	Si [%]	C [%]	O [%]	Si [%]
**Min.**	58.14	30.86	2.46	53.37	14.06	30.51
**Max.**	66.25	38.52	3.49	54.45	15.04	32.54
**Average**	63.75	33.41	2.84	53.91	14.55	31.54
**S.D.**	2.91	1.61	0.31	0.76	0.69	1.45

**Table 5 polymers-16-01739-t005:** Content of C, O, and Si in the samples PLA/PCL/SiO_2_ 90/10/1 and 90/10/3.

Sample	PLA/PCL/SiO_2_ 90/10/1			PLA/PCL/SiO_2_ 90/10/3		
Element	C [%]	O [%]	Si [%]	C [%]	O 1 [%]	Si [%]
Area PLA/PCL 90/10 without protrusions						
Min.	60.73	33.49	0.36	61.93	25.5	0.79
Max.	66.02	38.56	0.87	72.06	36.52	4.45
Average	63.09	36.33	0.58	67.65	30.02	2.13
S.D.	1.92	1.83	0.21	2.87	3.31	1.23
Protrusions—SiO_2_ clusters						
Min.	58.19	31.35	1.31	62.49	25.61	1.65
Max.	67.29	40.50	4.52	71.84	34.58	6.38
Average	63.35	33.88	2.76	67.09	29.25	3.66
S.D.	2.78	3.18	1.16	3.03	2.65	1.21

**Table 6 polymers-16-01739-t006:** Content of C, O, and Si in the samples PLA/PCL/SiO_2_ 70/30/1 and 70/30/3.

Sample	PLA/PCL/SiO_2_ 70/30/1			PLA/PCL/SiO_2_ 70/30/3		
Element	C [%]	O [%]	Si [%]	C [%]	O [%]	Si [%]
PLA matrix						
Min.	60.05	31.69	1.09	58.85	34.68	1.40
Max.	65.55	38.86	3.14	63.78	39.75	2.73
Average	62.35	35.96	1.69	61.24	36.81	1.96
S.D.	2.02	2.42	0.67	1.97	2.25	0.57
PCL component						
Min.	64.42	24.43	1.38	57.96	31.18	1.50
Max.	71.62	32.31	4.04	64.12	40.54	5.23
Average	69.30	27.90	2.80	61.12	36.43	2.45
S.D.	2.35	2.77	0.88	2.45	3.47	1.29
Protrusions—SiO_2_ clusters						
Min.	42.43	26.02	2.23	/	/	/
Max.	67.30	50.68	17.18	/	/	/
Average	55.26	34.13	10.61	/	/	/
S.D.	6.90	7.36	4.13	/	/	/

**Table 7 polymers-16-01739-t007:** Content of C, O, and Si in the samples PLA/PCL/SiO_2_ 50/50/1 and 50/50/3.

Sample	PLA/PCL/SiO_2_ 50/50/1			PLA/PCL/SiO_2_ 50/50/3		
Element	C [%]	O [%]	Si [%]	C [%]	O [%]	Si [%]
PLA matrix						
Min.	61.83	33.50	1.81	61.93	25.50	0.79
Max.	64.66	36.36	2.01	72.06	36.52	4.45
Average	62.84	35.26	1.90	67.65	30.02	2.13
S.D.	1.14	1.12	0.08	2.87	3.31	1.23
PCL component						
Min.	66.32	28.75	1.96	62.49	25.61	1.65
Max.	69.20	30.08	3.60	71.84	34.58	6.38
Average	67.84	29.47	2.69	67.09	29.25	3.66
S.D.	1.32	0.63	0.80	3.03	2.65	1.21

## Data Availability

The data presented in this study are available on request from the corresponding author due to privacy and contractual limit.

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
