# Peer review of "Quantitative Analysis of Morphology and Surface Properties of Poly(lactic acid)/Poly(ε-caprolactone)/Hydrophilic Nano-Silica Blends"

_polymers, 2024, doi:10.3390/polym16121739_

Round 1
Reviewer 1 Report
Comments and Suggestions for Authors
The article is devoted to the study of the morphology and composition of a polymer substance consisting of a mixture of two polymers PLA and PСL in different concentrations with the addition of silicon oxide nanoparticles.
The article is quite well designed, the material is presented logically. The main complaint about the presented work: an extremely weak justification for the prospect of using the voiced materials in practice. Examples are given of the use of each of the components, for example, as a non-resorbable material, but at the same time, what is the advantage of the composite being studied remains unclear to me. From this misunderstanding, the unreasonable goal of the work is further built: what do we want to get as a result.
Despite the overall strange impression of the work done, there are specific shortcomings, including those of a design nature::
1. All images obtained on SEM must be brought to a general form, eliminating the inscriptions on the images, as well as placing one type of marker on each one.
2. Specify the purpose of the work: not research, but what product do the authors want to get as a result, with what properties?
3. Correct the phase ratio error: 90/20 - 90/10 in Fig. 4b
4. Enter the descriptions of arrows and icons in the photo in the captions for the pictures. This applies to almost all SEM images. Remove extra characters.
5. Explain as a result of what influences Fig. 6a was obtained and why this insignificant “result” is presented at all? I recommend removing it.
6. It is necessary to combine Fig. 6b and Fig. 7, since we are talking about the geometry of the experiment.
7. An explanation is required of which domain (linear) we are talking about in Fig. 7? This assertion appears unreliable and unproven. In general, the results of the analysis of light elements by EDS raise significant doubts, especially with regard to carbon. Element concentration versus coordinate curves are noise and cannot be interpreted as true concentrations. Verification of this result by other methods is required.
Reviewer 2 Report
Comments and Suggestions for Authors
The aim of this study was a quantitative analysis of the morphology, the distribution of components and surface/interfacial properties in poly(lactic acid)(PLA), poly(ε-caprolactone (PCL) and nano-silica (SiO2) blends. This research aimed to improve understanding of how PLA, PCL, and nano-SiO2 interact, resulting in the specific morphology and surface properties of the blends. The prepared samples were characterized using SEM, EDS mapping, water contact angle measurements, surface free energy calculation, adhesion parameter measurements, and FTIR-ATR spectroscopy. The results showed that the addition of SiO2 nanoparticles led to an increase in the contact angle of water, making the surface more hydrophobic. SEM images of the blends showed that increasing the PCL content reduced the size of spherical PCL elements in the blends. FTIR-ATR analysis showed that hydrophilic SiO2 nanoparticles showed a mild effect on the reduction of the water bound by hygroscopic PLA. Shifts in the bands representing the crystalline phase of PLA indicated that nanoparticles influence the structure ordering of PLA in the blend with equal portions of PLA and PCL.
The paper is interesting and the discussion is well written. The results are nicely addresed. The paper can be accepted after minor revision. The novelty aspect of the study is good presented.
Minor issues:
1. The authors should discuss the potential applications of these materials.
2. Spectroscopic and morphological results should be correlated.
Reviewer 3 Report
Comments and Suggestions for Authors
The submitted work aims to improve understanding of how PLA, PCL, and nano-SiO2 interact, resulting in the specific morphology and surface properties of the blends. In this work, samples were produced by varying the concentration of all three components. They were analyzed using SEM, EDS mapping, water contact angle measurements, surface free energy calculation, adhesion parameter measurements, and FTIR-ATR spectroscopy. I think the results are exciting and were verified by the required characterizations. However, there is still a lack of necessary information that must be addressed before publication.
· Please add the name of the component in the abstract.
· In the keywords, add the full name, not only the abbreviation. In addition, In the first use, also add the full name, then use the abbreviations in the main manuscript
· quantitative analysis of their morphology improves, also add the results in the abstract.
· The introduction is very long it is a little short, and Conway's main background of the research, for example, lines 47-50, are not so crucial for the study introduction.
· There are various methods for producing and adjusting the properties of the resulting mixture to the intended application. Please add the method named and also cite the required references.
· Why do blends with 0% of SiO2 nanoparticles and with PCL in the samples display a lower water contact angle than pure PLA, just because of more hydrophobia or any other reason?
· Why the change in the proportion of PCL in the mixture did not significantly affect the change in the total surface free energy of the samples?
· The conclusion is too long. I think it should only be with the main obtained results it needs to be improved.
Final remarks: I think it can be published in polymers after addressing these comments.
Round 2
Reviewer 1 Report
Comments and Suggestions for Authors
Dear authors! I see that you have significantly improved the article and it can already be read. However, some shortcomings require additional improvement:
1. line 169: it is better to replace it with 60.
2. Too many images, often duplicating each other in meaning. I recommend reducing the number of repeated shots. There is no vertical gradation: it is not clear which samples each row belongs to. Only the columns are numbered. Line 292.. why all the samples? It will be more helpful to the reader if you identify the differences.
3. Bring the captions to the SEM pictures into uniformity. It is enough to leave a clearly readable marker mark, the same in all images: Figures 6,7,8,9,10,11.
